# The Dual Role of Neutrophil Extracellular Traps (NETs) in Sepsis and Ischemia-Reperfusion Injury: Comparative Analysis across Murine Models

**DOI:** 10.3390/ijms25073787

**Published:** 2024-03-28

**Authors:** Antonia Kiwit, Yuqing Lu, Moritz Lenz, Jasmin Knopf, Christoph Mohr, Yannick Ledermann, Michaela Klinke-Petrowsky, Laia Pagerols Raluy, Konrad Reinshagen, Martin Herrmann, Michael Boettcher, Julia Elrod

**Affiliations:** 1Department of Pediatric Surgery, University Medical Center Hamburg-Eppendorf, Martini Strasse 52, 20246 Hamburg, Germany; 2Department of Pediatric Surgery, University Medical Center Mannheim, Heidelberg University, Theodor-Kutzer-Ufer 1-3, 68167 Mannheim, Germany; 3Department of Internal Medicine 3—Rheumatology and Immunology, Friedrich-Alexander-Universität Erlangen-Nürnberg (FAU), Universitätsklinikum Erlangen, Ulmenweg 18, 91054 Erlangen, Germany; 4Deutsches Zentrum für Immuntherapie, Friedrich-Alexander-Universität Erlangen-Nürnberg (FAU), Universitätsklinikum Erlangen, Ulmenweg 18, 91054 Erlangen, Germany

**Keywords:** sepsis, neutrophil extracellular traps (NETs), midgut volvulus, cecal ligation and puncture (CLP), lipopolysaccharide (LPS)

## Abstract

A better understanding of the function of neutrophil extracellular traps (NETs) may facilitate the development of interventions for sepsis. The study aims to investigate the formation and degradation of NETs in three murine sepsis models and to analyze the production of reactive oxygen species (ROS) during NET formation. Murine sepsis was induced by midgut volvulus (720° for 15 min), cecal ligation and puncture (CLP), or the application of lipopolysaccharide (LPS) (10 mg/kg body weight i.p.). NET formation and degradation was modulated using mice that were genetically deficient for *peptidyl arginine deiminase-4 (PAD4*-KO) or *DNase1* and *1L3 (DNase1/1L3*-DKO). After 48 h, mice were killed. Plasma levels of circulating free DNA (cfDNA) and neutrophil elastase (NE) were quantified to assess NET formation and degradation. Plasma deoxyribonuclease1 (DNase1) protein levels, as well as tissue malondialdehyde (MDA) activity and glutathione peroxidase (GPx) activity, were quantified. DNase1 and DNase1L3 in liver, intestine, spleen, and lung tissues were assessed. The applied sepsis models resulted in a simultaneous increase in NET formation and oxidative stress. NET formation and survival differed in the three models. In contrast to LPS and Volvulus, CLP-induced sepsis showed a decreased and increased 48 h survival in *PAD4*-KO and *DNase1/1L3*-DKO mice, when compared to WT mice, respectively. *PAD4*-KO mice showed decreased formation of NETs and ROS, while *DNase1/1L3*-DKO mice with impaired NET degradation accumulated ROS and chronicled the septic state. The findings indicate a dual role for NET formation and degradation in sepsis and ischemia-reperfusion (I/R) injury: NETs seem to exhibit a protective capacity in certain sepsis paradigms (CLP model), whereas, collectively, they seem to contribute adversely to scenarios where sepsis is combined with ischemia-reperfusion (volvulus).

## 1. Introduction

Sepsis, a serious medical condition, results from a dysregulated host immune response. It leads to life-threatening organ dysfunction and remains a significant threat to the health and well-being of children [1,2]. In 2017, sepsis affected 48.9 million people worldwide, with a mortality of 11 million [3]. Pediatric sepsis is a multifaceted condition resulting from infection by various bacteria, viruses, fungi, and parasites, or their toxic products [4]. Moreover, following trauma or burns, tissue damage can induce a comparable hyperinflammatory response known as sterile sepsis, mediated by factors such as uric acid, High mobility group box 1(HMGB1), or mitochondrial DNA, among others [5].

Neutrophils play important roles in infection-mediated sepsis. They migrate from the circulation into infected tissues and form neutrophil extracellular traps (NETs) [5,6]. NETs are complex structures composed of DNA, histones, and antimicrobial proteins. They are released by activated neutrophils as a part of their pathogen defense mechanisms [7,8]. NETs trap and kill pathogens and, thus, prevent the spread of infection. However, excessive NET formation can cause tissue damage, coagulopathy, and eventual distant organ failure [8,9,10]. Elevated levels of neutrophil extracellular traps (NETs) have the potential to detrimentally impact endothelial function, resulting in compromised blood flow and tissue ischemia, thereby exacerbating the pathophysiological manifestations of sepsis [11]. Endothelia are affected by NETs in a proinflammatory and proangiogenic manner. This dysregulates and further destabilizes the immune system [12,13]. Activated endothelial cells increase glycolysis, further promoting inflammation and oxidative stress [14]. Thus, NETs can trigger exaggerated inflammatory responses that lead to a cytokine storm and progression of sepsis [15]. Components of NETs can also give rise to pathological immunothrombi, non-canonically triggering coagulation [16,17,18] to occlude vessels, particularly capillaries, thereby precipitating additional organ damage [19]. Neutrophils are also crucial for the pathogenesis of I/R [20]. NETs, reactive oxygen species (ROS), and immune cell recruitment form a self-perpetuating vicious circle of hyperinflammation [21,22].

Chromatin decondensation allows NETs to expand and to form larger aggregates [23]. It is driven by the deamination of the histones H3 and H4 by peptidyl arginine deiminase 4 (*PAD4*) [24]. The degradation of NETs is initiated by Plasma deoxyribonuclease1 (DNase1) and DNase1L3. Secreted DNase1 cleaves the NET-borne DNA on sites not covered by proteins and produces small NET fragments, referred to as NET degradation products [25]. DNase1L3, also known as DNase γ, is secreted mostly by immune cells and targets DNA-protein complexes such as nucleosomes [19]. DNase1L3 clears NETs in blood vessels in the course of sepsis or sterile neutrophilia [26]. Moreover, DNase1L3 is highly expressed in plasmacytoid dendritic cells [27] and other myeloid immune cells. It can be found in the circulation [28] and in the vicinity of NETs. Both enzymes contribute to the degradation of NETs by cleaving the DNA backbone, rendering them susceptible to further degradation by other nucleases [29].

Here we investigate the contribution of NETs to the development and mortality of LPS, CLP, or volvulus-driven sepsis models. We employed *PAD4*-KO and *DNase1/DNase1L3*-DKO mice to mimic the inhibition of NET formation and degradation, respectively [29]. The midgut volvulus model serves as a physiological and highly reproducible model of ischemia-reperfusion-induced sepsis [21]. Our results provide further insight into the mechanisms of sepsis and open up new avenues for clinical treatment.

## 2. Results

### 2.1. Survival in the Septic Models Depended on the Modulation of Neutrophil Extracellular Trap Levels

The survival rate of the three septic models depended on the level of circulating NETs; specifically, the knockdown of *DNase1* and/or *DNase1L3* prevented the degradation of NETs and, thus, increased their accumulation. This exacerbated injury and decreased survival rates in volvulus and LPS-induced sepsis; however, statistical significance was not reached. In contrast, knockdown of the *PAD4* gene prevented NET expansion and aggregation, thus increasing survival in the volvulus and LPS-model (see Figure 1A,B). In the case of CLP, the knockouts had the opposite effect. *DNase1/1L3* and *PAD4*-KO mice showed a slightly, but not significantly, increased and non-significantly decreased survival rate compared to wild-type mice (Figure 1C).

### 2.2. Plasma Levels of Cell-Free DNA and DNase1

In all three WT models, cfDNA was not increased as compared to the controls. However, cfDNA increased in *DNase1/1L3*-DKO mice undergoing sepsis. Notably, following the application of volvulus, a moderate increase in cfDNA was observed in *PAD4*-KO mice compared to WT mice (refer to Figure 2A–C).

There was a significant increase in DNase1 expression in wild-type mice exposed to all three types of sepsis, compared to the controls. This indicates an increase in NET formation. In contrast, *PAD4-KO* mice showing diminished NET formation had lower plasma DNase levels than WT mice subjected to volvulus. The absence of DNases in *DNase1/1L3*-DKO mice indicates successful gene editing (see Figure 2D–F). The amount of NE in the blood parallels the quantity of NETs released and is not affected by the degradation of DNA strands within NETs [30]. The ELISA data, along with the expression pattern of NE, exhibited some similarities to those of cfDNA (refer to Figure 2G–I).

### 2.3. Tissue Expression of DNase1 and DNase1L3

We analyzed the expression of DNase1 and DNase1L3, two types of deoxyribonucleases that play an important role in the degradation of DNA [31], in the intestine (Figure 3A), liver (Figure 3B), and lung (Figure 3C) in representative images of immunofluorescence staining for each model. Control animals are shown in Figure 3D. Both enzymes were upregulated to varying degrees in all sepsis models. Under sepsis conditions, the liver, lung, and intestines of *PAD4*-KO mice showed a somewhat increased expression of DNase1 and DNase1L3, when compared to WT mice in response to sepsis. At the same time, *DNase1/1L3*-DKO mice showed no expression of DNase1 and DNase1L3, indicating, again, successful gene editing.

### 2.4. Oxidative Stress and Antioxidative Capacity

Malondialdehyde (MDA) is a highly reactive toxic compound formed as a by-product of lipid peroxidation when unsaturated fatty acids are exposed to ROS [32]. MDA serves as a biomarker of lipid peroxidation and oxidative stress, and its levels have been used to monitor oxidative damage in various pathological conditions [33]. We observed reduced oxidative stress damage in all tissues of *PAD4*-KO and *DNase1/1L3*-DKO, see Figure 4. ROS showed an inverse appearance, possibly due to impaired NET degradation during sepsis. In the volvulus model, the loss of *DNase1/1L3* had a more pronounced effect on ROS compared to *PAD4*-KO. Furthermore, we investigated the role of glutathione peroxidase (GPx), a family of enzymes crucial in protecting cells from oxidative stress, see Figure 5. They catalyze the reduction of hydrogen peroxide and organic peroxides using reduced glutathione (GSH) as a co-substrate [34]. We found increased GPx levels in the lungs of *PAD4*-KO compared to WT mice. This may be attributed to a reduction in *PAD4*-dependent NET formation, leading to increased ROS formation [35].

## 3. Discussion

The objective of this study was twofold: (I) to examine the formation and degradation processes of neutrophil extracellular traps (NETs) in three widely employed murine models of sepsis, and (II) to deepen our understanding of the influence of NETs on the progression and outcomes of sepsis.

Interestingly, the analysis of the 48 h survival rates showed that the number of NETs had opposite effects in the three models of sepsis. NETs seemed to display an overall beneficial effect on survival in the CLP model; however, they showed a somewhat reduced survival rate in the volvulus and LPS models. The favorable effect in the CLP model might be explained by the high bacterial load and the very high focal inflammatory state after intestinal perforation [21,36,37,38]. NETs are required to trap and phagocytose these pathogens. This mechanism plays a crucial role in shielding the organism from bacterial dissemination, inflammation, and distant injury [39]. In contrast, the diminished NET formation due to *PAD4*-KO could lead to diminished overwhelming inflammation [40]. This in turn could potentially decrease NET-induced platelet adhesion, activation, and aggregation, which are known precursors to thrombus formation, as described by Fuchs et al. [41]. In sepsis models where thrombus formation is critically harmful, such as in the volvulus model, or where bacterial antigens initiate pathology rather than the bacteria themselves, as in the LPS model, the role of NETs in pathogen capture might be less critical. Therefore, the absence of *PAD4*, by reducing NET formation, could lead to less thrombus formation and less intense inflammation, thereby offering a protective advantage.

Our findings showcase the complex role of neutrophil extracellular traps (NETs) in sepsis. This study was conducted using three distinct sepsis models in *PAD4* and *DNase1/1L3*-DKO mice. To the best of our knowledge, such a systematic comparative analysis has not been performed before. The pathophysiology of each sepsis model used in this study varies significantly, as described recently.

The cecal ligation and puncture (CLP) model is a commonly used sepsis model, in which holes are made in the ligated but otherwise healthy cecum. Different parameters of CLP, such as the location of the ligation and the size and number of the holes can be selected to adjust for disease severity, leading to varying mortality rates [42]. This method causes mainly three types of damage to the host: first, tissue injury due to surgery; second, tissue ischemia resulting from the tied-off cecum; and third, a polymicrobial infection caused by the leakage of fecal matter following needle perforations [43]. The spread of bacteria from the intestine is a key driver of sepsis and fatal outcomes [44]. This is where NETs are beneficial overall; they trap pathogens, limit bacterial dissemination, control disease progression, and reduce mortality. However, previous studies examining the effect of DNase application in the CLP model have indicated that the timing of its administration can have opposing effects: early application might increase pro-inflammatory cytokines and exacerbate organ damage, whereas later application may result in anti-inflammatory, organ-protective conditions and reduced bacterial dissemination and survival [45]. These findings underscore the intricate role of NETs in sepsis, emphasizing the need for more thorough research to enhance our understanding of this area [46].

In contrast to CLP, in LPS-induced sepsis, disseminated intravascular coagulation (DIC) and a cytokine storm are the primary driving factors [47]. Here, NET formation most likely seems to exacerbate endothelial damage and precipitate DIC. This may explain the decreased survival when NET levels increase, as seen in *DNase1/1L3*-DKO mice. It is important to note that the pathophysiology of this model predominantly involves the cytokine storm initiated by LPS rather than the immune response to active bacteria. Additionally, the LPS model reveals secondary bacterial translocation [21], which is likely due to a sequentially impaired intestinal barrier, a phenomenon present in numerous inflammatory diseases [48]. As demonstrated by Sun et al. [49], NETs may contribute to the deterioration of intestinal barrier function, yet, concurrently, bacterial clearance might be crucial for survival at this stage, once more highlighting the complex and potentially dual role of NETs in sepsis.

In volvulus, ischemia-reperfusion injury is caused by the torsion and subsequent detorsion of the intestine, leading to inflammation and increased intestinal permeability, which is a primary mechanism in its pathogenesis [50]. Additionally, in advanced stages, significant necrosis and inflammation can result in perforation and the development of gangrene [50]. The volvulus sepsis model may therefore offer a more comprehensive representation of human sepsis, as it facilitates bacterial migration from the damaged mucosal layer into the vascular system. This context underlines the potential for an accumulation of NETs to aggravate endothelial damage [51], possibly accelerating the progression of sepsis and leading to increased mortality, as demonstrated in the current study.

In the present study, NET load was quantified inter alia by the measurement of the levels of circulating cfDNA and NE. Associations between the levels of cfDNA, inflammation, and coagulation have previously been described in various pathological conditions [52].

During sepsis, neutrophils abundantly infiltrate the affected tissue, as well as remote organs like the liver and lung. This process is accompanied by NET formation releasing decondensed chromatin and, thus, generating variable amounts of cfDNA [10]. The present study consistently showed a trend of increased cfDNA in all models. Moreover, the cfDNA levels were further increased in *DNase1/1L3*-DKO and *PAD4*-KO mice. These findings suggest a shared mechanism of disease progression or response in all models.

Two days post-induction, *DNase1/1L3*-DKO mice exhibited the highest levels of circulating free DNA (cfDNA) compared to all other groups across all three sepsis models. This might be due to the loss of the DNases, which degrade extracellular DNA [53]. The loss of *PAD4* results in a reduced protein citrullination [54]. Interestingly, *PAD4*-KO mice also demonstrated higher levels of cfDNA than WT mice in the sepsis models. This is somewhat unexpected, since a *PAD4* inhibition previously reportedly reduced the formation of NETs [55]. However, in other reports, *PAD4*-KO mice still formed NETs, but these are smaller, since *PAD4* is required for chromatin decondensation. Furthermore, the *PAD4*-independent NETs showed a reduced tendency to aggregate and to form ductal plugs in the pancreas [56]. Alternatively, NET-independent generation of cfDNA may occur during sepsis. Both late apoptosis and necrosis may add to the production of cfDNA during sepsis [57,58]. The high DNase concentrations are most likely responsible for the low levels of circulating cfDNA. NE, a prerequisite for NET formation, degrades the cytoskeleton and disrupts the cell membrane upon activation of MPO, which leads to the release of NETs [59]. In this context, ELISA measurements of circulating neutrophil elastase (NE) levels and the corresponding expression trends demonstrated a certain degree of consistency with those of cfDNA in all groups.

DNase1 and DNase1L3 are crucial enzymes for the degradation of NETs [60]. Both endonucleases cleave the DNA backbone, rendering them susceptible to further degradation by other nucleases [61]. Our experiments revealed the increased expression of DNase1 and DNase1L3 in *PAD4*-KO, compared to WT mice, in almost all three sepsis models. This increase may be due to the reduced tendency of NETs from condensed chromatin to aggregate and to form indigestible clumps in the absence of *PAD4* [62]. *PAD4*-dependent NETs are then more easily cleaved by nucleases. Interestingly, this change was not observed in the intestine of the volvulus group, possibly due to pathological alterations characterized by ischemia/reperfusion (I/R) and sepsis resulting from the volvulus model. *PAD4* gene knockdown did not completely inhibit intestinal I/R in the volvulus model [37]. 1 We hypothesize that this finding might be related to DNase1L3 sensitivity to proteolysis by plasmin, whereas DNase1 seems to be resistant to plasmin, possibly due to its di-*N*-glycosylation, which is not present in DNase1L3 [63].

Some pathways of NET formation depend on ROS production. The oxidative burst generates ROS, activates enzymes that break down nuclear and cytoplasmic components, and augments NET release [64]. While ROS foster NET formation and pathogen defense, excessive ROS has detrimental effects for many tissues [65]. In the present experiments, the loss of DNase1 and DNase1L3 impaired NET degradation and caused NETs to accumulate. ROS production showed an opposite trend, which might be explained by impaired DNase1 and DNase1L3 degradation during sepsis, a phenomenon previously described by others [66]. Another potential explanation for the accumulation of NETs and the reduction of ROS is that the loss of DNase1 and DNase1L3 impairs their protective impact on the pathogens during sepsis. This, in turn, accelerates DNA and tissue damage as well as oxidative stress [26]. Consequently, the sepsis models exhibit ROS-depleted states, displaying reduced MDA levels at 48 h. In the volvulus model, the suppression of DNase1 and DNase1L3 seemed to exert a more pronounced impact on ROS compared to the *PAD4*-KO. We argue that the volvulus model represents a combined ischemia/reperfusion (I/R) and sepsis model. I/R exacerbates ROS-induced damage within the organism, resulting in a more profound depletion state after 48 h. Moreover, the present experiments suggest that *DNase1/1L3*-DKO mice have a diminished antioxidant capacity in lung, liver, and intestinal tissues under septic conditions when compared with WT mice. This could further support our initial hypothesis that *DNase1* and *DNase1L3* suppression might result in accelerated NET and ROS-induced accumulation of DNA damage, possibly depleting the protective antioxidant capacity.

Due to the tissue-specific distribution of GPx isoforms, GPx1 is the predominant isoform in lung tissues and is crucially important for protecting cells from oxidative damage [34]. GPx1 plays a role in the overall recovery of cells after oxidative stress [61]. GPx1 uses glutathione (GSH) to reduce ROS. Consistent with our observations, the oxidation product glutathione disulfide (GSSG) is then reconverted to GSH by glutathione reductase [67]. In contrast, GPx2 is primarily expressed in the gastrointestinal tract and its primary function is the protection of the intestinal epithelia against oxidative damage [68]. This distinction might explain the observed reduction in GPx expression within the intestines of *PAD4*-KO mice, compared to WT mice.

### Limitations

The study has several shortcomings. For an enhanced elucidation of NETs’ function in sepsis, a longitudinal study with staggered endpoints would be beneficial. Animal welfare regulations necessitated the sole utilization of the 48 h mark, a timepoint selected to affirm sepsis establishment while preserving the viability of statistical evaluations, which would be compromised at a later endpoint due to escalating mortality rates. 

Moreover, certain aspects, such as the immunofluorescence (IF) staining of DNase1 and DNase1L3 in tissue samples quantification or the inclusion of mRNA data, could potentially enhance and refine the interpretations of our findings.

## 4. Conclusions

The intricate interplay between sepsis and NETs delineated in this study emphasizes the delicate equilibrium and the dual role played by NETs in the context of sepsis: NETs demonstrate a protective function in the CLP model by limiting bacterial dissemination. In contrast, in the volvulus and LPS models, an imbalance in NET production or clearance exacerbates inflammation and possibly thrombosis, adversely affecting survival. A comprehensive understanding of this interplay might be beneficial for the development of novel therapeutic strategies to treat sepsis and reduce morbidity and mortality. Selective immunomodulation, which includes targeting NET formation and restoring their clearance, potentially holds promise for enhancing patient outcomes in sepsis.

## 5. Materials and Methods

### 5.1. Study Design

Ethical approval was obtained from the Hamburg State Administration for animal research (TVA 100/17; date of approval: 10 April 2018). A total of 79 six to eight-week-old mice were utilized, all having the same genetic background (C57BL/6). *PAD4*-KO mice (B6.Cg-Padi4tm1.1Kmow/J) were purchased from Jackson Laboratories. *DNase1/1L3*-DKO mice were generously provided by Prof. Winkler, Department of Biology, Friedrich-Alexander University Erlangen-Nuremberg, Germany, after breeding *DNase1*-KO mice with *DNase1L3*-KO mice [36,69].

All procedures performed and parameters applied within the animal facility complied with the German guide for the care and use of laboratory animals.

### 5.2. Animal Procedures

All mice were anesthetized with 5% isoflurane (Baxter, Unterschleißheim, Germany) and maintained throughout the procedure, with 2.5% isoflurane gas delivered via a face mask. Betaisodona (Mundipharma GmbH, Frankfurt am Main, Germany) was used for preoperative antisepsis. For analgesia, all mice received buprenorphine (Reckitt Benckiser, Mannheim, Germany) at a dose of 0.1 mg/kg body weight, subcutaneously, 30 min preoperatively. Briefly, for the control condition, mice were positioned in a dorsal decubitus posture and underwent a median laparotomy to expose the intestine for 1 min. The abdomen was then closed using simple interrupted sutures (*n* = 5). For the volvulus procedure, following a median laparotomy, the small intestine was exposed and twisted by 720° for 15 min before being returned to its original position. The abdomen was closed using simple interrupted sutures (8× WT, 8× *PAD4*-KO, 8× *DNase*-DKO; *n* total = 24). For the CLP procedure, after a median laparotomy, the cecum was ligated at the midpoint and punctured twice in the distal portion using a 19 G cannula. The abdomen was closed using simple interrupted sutures (11× WT, 8× *PAD4*-KO, 8× *DNase*-DKO; *n* total = 27). For the LPS procedure, LPS-EB Ultrapure (LPS-EB Ultrapure, Invivogen, San Diego, CA, USA) was intraperitoneally injected at a dose of 10 mg/kg body weight (1 mg/1 mL LPS) (11× WT, 8× *PAD4*-KO, 8× *DNase*-DKO; *n* total = 27). To approximate neutrophil counts in mice to human conditions, all mice received 250 µg/kg body weight of Lenograstim (Granocyte 34, G-CSF, Chugai Pharmaceutical, Tokyo, Japan) subcutaneously after the procedure, and an additional dose was given after 24 h if not killed at that time point. This procedure, as shown previously [70], leads to a significant increase in the neutrophil count, resulting in a better reflection of human neutrophil levels, in which the blood is rich in neutrophils, while the blood of mice shows a strong preponderance of lymphocytes [71,72]. Scoring of the mice was carried out as mandated by the ethics committee at regular intervals, considering both an adapted Murine Sepsis Score and tailored criteria. For details regarding the animal procedures, please refer to our previous study.

#### 5.2.1. Circulating Free Deoxyribonucleic Acid (cfDNA)

Plasma cfDNA levels were quantified as previously described [27], using a SytoxGreen fluorescence-based assay.

#### 5.2.2. Deoxyribonuclease I (DNase1)

Plasma DNase1 protein levels were determined using the Mouse Deoxyribonuclease 1 (DNase1) ELISA kit (MyBioSource, San Diego, CA, USA), following the manufacturer’s instructions.

#### 5.2.3. Neutrophil Elastase (NE) Enzyme-Linked Immunosorbent Assay (ELISA)

Plasma NE levels were quantified using the elastase neutrophil expressed (ELANE) ELISA kit (Boster Biological Technology, Pleasanton, CA, USA), following the manufacturer’s instructions.

#### 5.2.4. Immunofluorescence Staining

Separate staining for DNase1 and DNase1L3 was performed using the protocol described previously [73]. Sections were incubated overnight at 4 °C with DNase1-polyclonal antibody (BS-7651R, Bioss, Woburn, MA, USA, 0.01 µg/mL) or DNase1L3-polyclonal antibody (BS-7653R, Bioss, Woburn, MA, USA, 0.01 µg/mL). The next day, sections were incubated for 30 min at room temperature with a donkey anti-rabbit-IgG Alexa Fluor 647 (AB150075, Abcam, Cambridge, UK) and DAPI (Invitrogen, Grand Island, NY, USA). Images were captured using a fluorescence microscope (KEYENCE BZ-9000, Keyence Corporation, Osaka, Japan). Exposure time was calibrated using the isotype control antibody (AB37415, Abcam, Cambridge, UK), with adjustments made to achieve a negative fluorescence signal in the control. Three to five samples were inspected microscopically, and one representative image was taken per condition.

#### 5.2.5. Glutathione Peroxidase (GPx) Assay

Tissue GPx activity (liver, lung, and small intestine), a marker of systemic antioxidant status, was measured using a glutathione peroxidase assay kit (Cayman, Ann Arbor, MI, USA) according to the manufacturer’s instructions.

#### 5.2.6. Malondialdehyde (MDA) Assay

Tissue MDA activity (liver, lung, spleen), a marker for lipid peroxidation, was measured using the MDA assay kit (Sigma Aldrich, St. Louis, MO, USA), according to the manufacturer’s instructions.

### 5.3. Statistics

Data were analyzed using GraphPad Prism 9.4.1 (GraphPad, San Diego, CA, USA). The Mantel–Cox (log-rank) test was applied to compare the survival distributions among the groups; in all other cases, an ordinary one-way ANOVA was used for multiple comparisons. Data are presented as mean ± standard deviation (SD). The level of significance was set at 0.05 [74].

## Figures and Tables

**Figure 1 ijms-25-03787-f001:**
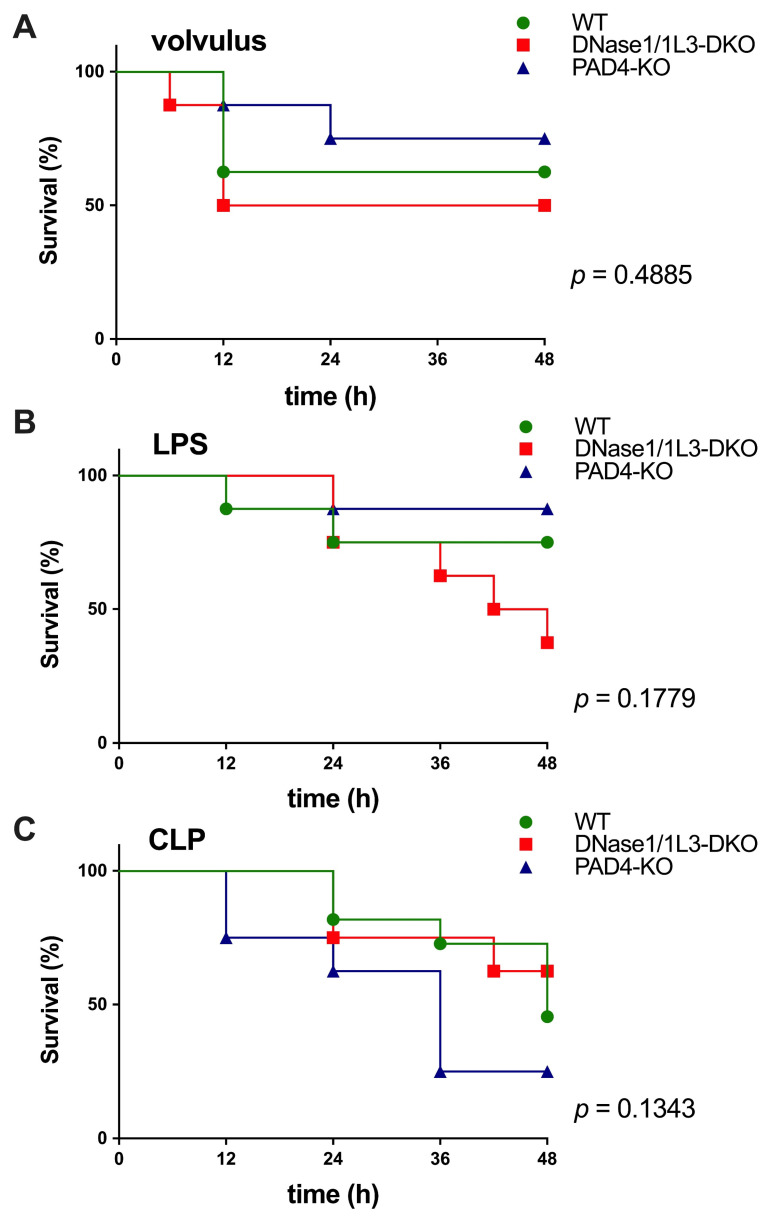
NETs differentially modulate the 24 h survival rate in sepsis induced by volvulus, LPS or CLP. (**A**) Survival rates for the volvulus model for WT (*n* = 8, 62.5%), *PAD4*-KO (*n* = 8, 75%), and *DNase1/1L3*-DKO (*n* = 8, 50%) groups at 48 h. (**B**) Survival rate for the LPS model for WT (*n* = 7, 71.4%), *PAD4*-KO (*n* = 8, 87.5%), and *DNase1/1L3*-DKO (*n* = 8, 37.5%) groups at 48 h. (**C**) Survival rates for the CLP model for WT (*n* = 11, 45.4%), *PAD4*-KO (*n* = 8, 25%), and *DNase1/1L3*-DKO (*n* = 8, 62.5%) groups at 48 h. The Mantel–Cox (log-rank) test was applied to compare the survival distributions among the groups. *p*-values are indicated above, showing that the findings were not significant.

**Figure 2 ijms-25-03787-f002:**
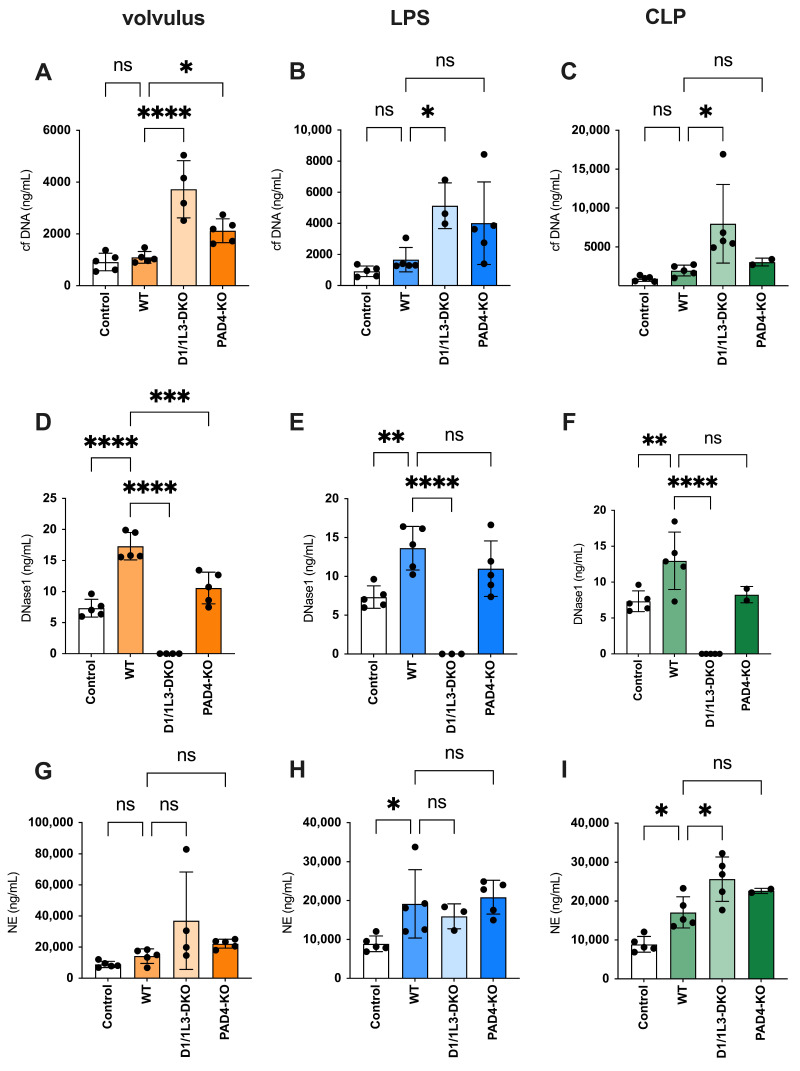
During sepsis, cfDNA, DNase1, and neutrophil elastase (NE) plasma levels increase. Plasma levels of the canonical NET markers cfDNA, DNase, and NE are increased in all three sepsis models. (**A**–**C**) The levels of cfDNA in volvulus (**A**), LPS (**B**), and CLP (**C**); (**D**–**F**) levels of DNase in volvulus (**D**), LPS (**E**), and CLP (**F**); NE plasma levels in volvulus (**G**), LPS (**H**), and CLP (**I**) 48 h after intervention. Ordinary one-way ANOVA (Analysis of Variance) was used. ns, not significant, * *p* < 0.05, ** *p* < 0.01, *** *p* < 0.001, **** *p* < 0.0001.

**Figure 3 ijms-25-03787-f003:**
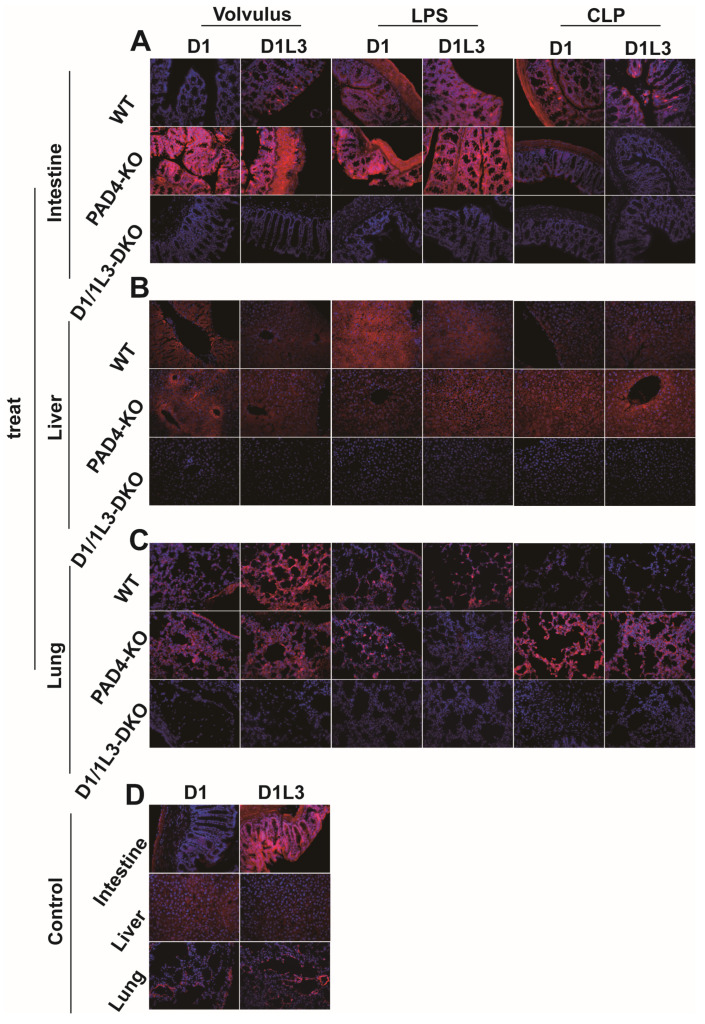
DNase1 and Dnase1L3 expression is increased in *PAD4*-KO mice. Representative merged images of immunofluorescence staining for DNase1 and DNase1L3 in the intestine, liver, and lung: (**A**) intestine, 40× magnification; (**B**) liver, 40× magnification; (**C**) lung, 40× magnification; (**D**) control: median laparotomy, exposure of the intestine for 1 min, closing the abdomen, no further manipulation. Magnification was 40×. DNA was stained with DAPI (blue). treat, treatment groups.

**Figure 4 ijms-25-03787-f004:**
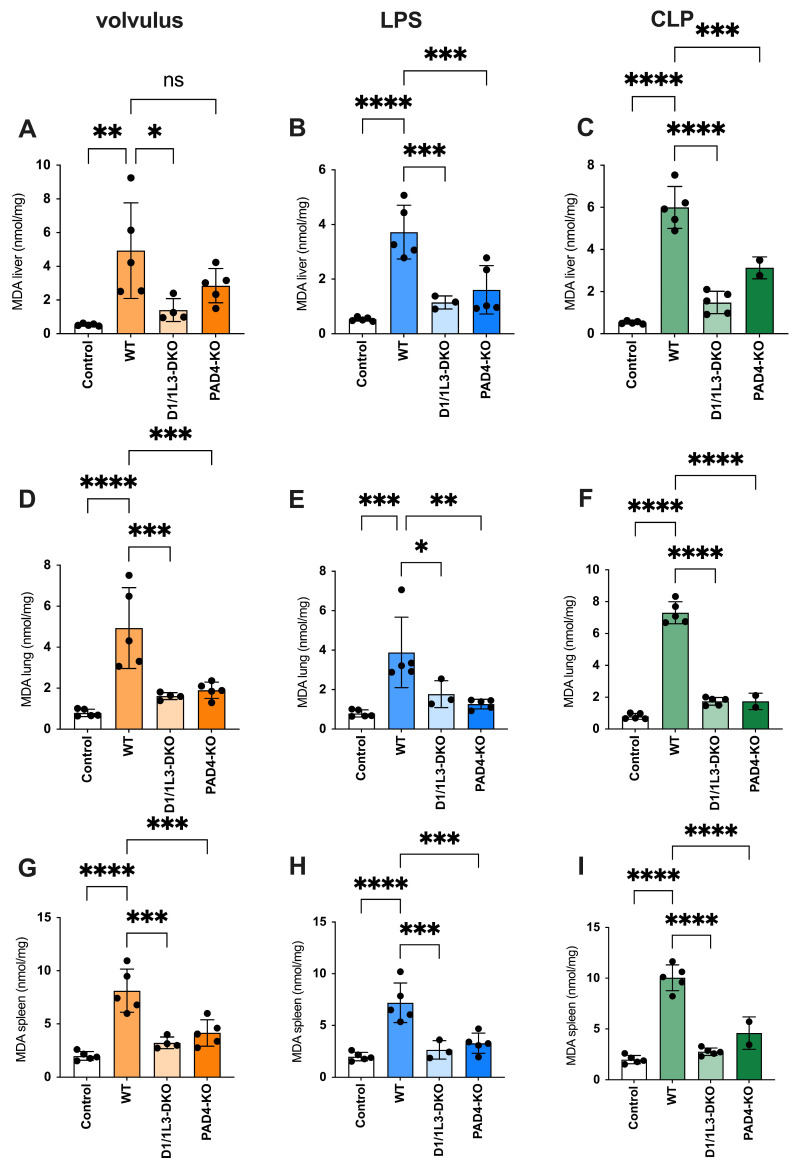
*PAD4*-KO and *DNase1/1L3*-DKO mice show reduced oxidative stress. MDA levels in liver, lung, and spleen (**A**,**D**,**G**) of volvulus, (**B**,**E**,**H**) LPS, and (**C**,**F**,**I**) CLP-induced sepsis, respectively. Ordinary one-way ANOVA was used. ns, not significant. * *p* < 0.05, ** *p* < 0.01, *** *p* < 0.001, **** *p* < 0.0001.

**Figure 5 ijms-25-03787-f005:**
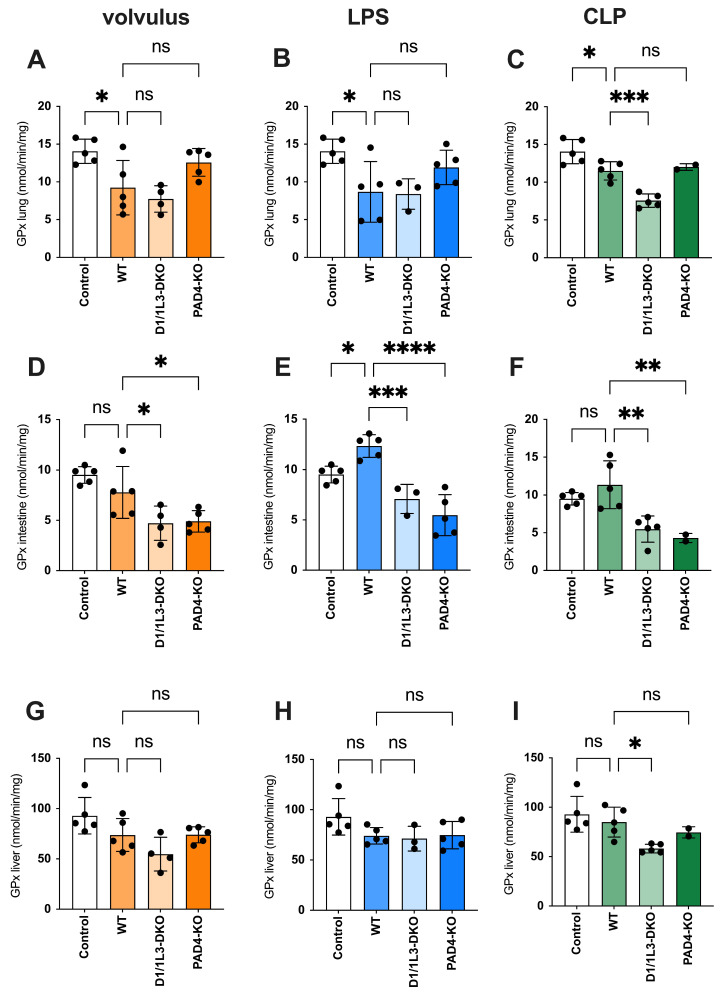
The antioxidant capacity is impaired in *DNase1/1L3*-DKO mice. GPx levels in lungs, intestines, and livers of (**A**,**D**,**G**) volvulus, (**B**,**E**,**H**) LPS, and (**C**,**F**,**I**) CLP-induced sepsis, respectively. Ordinary one-way ANOVA was used. ns, not significant * *p* < 0.05, ** *p* < 0.01, *** *p* < 0.001, **** *p* < 0.0001.

## Data Availability

The data presented in this study are available on request from the corresponding author.

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
