# Peer review of "The Dual Role of Neutrophil Extracellular Traps (NETs) in Sepsis and Ischemia-Reperfusion Injury: Comparative Analysis across Murine Models"

_ijms, 2024, doi:10.3390/ijms25073787_

Round 1

Reviewer 1 Report

Comments and Suggestions for Authors

This study investigated NETs in three murine sepsis models with the aim of determining factors that  control the formation and degradation of NETs. The formation of NETs was decreased in PAD4-KO mice and their degradation was impaired in DNase1/1L3-DKO mice. The results appear to be scientifically sound. The data are interesting. Some corrections are necessary to make the manuscript suitable for publication.

Major comments:

1. How were the knockout mice generated, or form where have they been obtained? Do these mice have any disease phenotypes prior to the treatments?

2. The graphs in figures 1, 4 and 5 and some of the graphs in figure 2 lack the units of measurement. Numbers without units are not acceptable.

3. The expression of DNase 1 and DNase 1L3 was determined by immunofluorescence labeling. Data on mRNA levels would be desirable. If obtaining data is difficult, is there literature / transcriptome data? What is the control in Figure 3? Explain it in the legend.

4. Some of the statements are not supported by data and should be toned down. For example, in the conclusions (Line 427) the authors write “A comprehensive understanding of this interplay is a conditio sine qua non for the development of novel therapeutic strategies to effectively treat sepsis and reduce morbidity and mortality.” The reviewer doubts that this is truly an essential condition any novel effective therapy.

5. The title of the manuscript is not suitable for an original article because it lacks information about the content or conclusions.

Minor comments

Please correct the list of affiliations.

Abstract:

Line 55: “Circulating deoxyribonuclease (DNase)” or “circulating cfDNA”?

Line 63: correct “chronicled”

The last two sentences of the Abstract should be improved.

Reviewer 2 Report

Comments and Suggestions for Authors

Kiwit et al. investigated the role of NETosis in three different models of sepsis. Furthermore, they induce sepsis in two genetically determined deficiencies for peptidyl arginine deiminase-4 (PAD4-KO) or DNase1 and 1L3 (DNase1/1L3-DKO).

Comments

1. The conclusion of the summary is not clear

2. The graphical summary appears to be incomplete and does not provide any conclusions from the study

3. The techniques used to assess NETosis are not very sensitive. Detection of MPO using flow cytometry or imaging might be suitable for NETosis assessment.

4. Bacterial load/count is not assessed between models in the manuscript but it is mentioned in the discussion. This would be advantageous if the authors had estimated the bacterial load.

5. Figure 3: Immunofluorescence assessment can be converted into quantitative values

6. Why PAD4 KO models are protected from volvulus and LPS models while CLP models are disease exacerbation. This should be explained in the discussion.

7. The conclusion is not clear and should be consistent with the results of the study

Reviewer 3 Report

Comments and Suggestions for Authors

The manuscript provides valuable insights into the role of neutrophil extracellular traps (NETs) in murine sepsis models. Authors are suggested to improve the quality of article by considering the following suggestions.

Suggestions:

1. The abstract lacks a concise summary of the methodology and key findings.

2. The number of animals used in each group and the rationale for the group sizes should be explicitly stated. This information is crucial for assessing the statistical power of the study.

3. Authors should clarify the rationale behind the choice of 48 hours as the endpoint for survival analysis.

4. Results are poorly stated. Authors should more focused of key findings/significant outcomes and avoid unnecessary repetition

5. Authors should ensure that the statistical analyses are appropriately described and applied, including the rationale for the chosen tests and corrections for multiple comparisons.

6. Discussion is comprehensive. Authors are suggested to consider a more structured approach to highlight the main findings by emphasizing the significance of the observed differences in NETs effects between the sepsis models.

7. Some statements lack references from the study data. Authors should provide evidence to substantiate claims particularly in the discussion of the different sepsis models.

8. Authors should address potential limitations of the study.

9. Authors should succinctly summarize the main findings in conclusion section and their implications for future research.

General Suggestions:

1. The manuscript requires thorough proofreading and language editing for clarity, grammar and scientific accuracy.

2. Authors should ensure consistent formatting style throughout the manuscript and correct grammatical and typographical errors.

Comments on the Quality of English Language

The manuscript requires thorough proofreading and language editing for clarity, grammar and scientific accuracy.

Round 2

Reviewer 1 Report

Comments and Suggestions for Authors

Thank you for the corrections. Two points should be corrected at the Proof stage:

Title: correction of "duale"

Affiliations: in the pdf of the manuscript, author names are shown again as part of the affiliation.

Comments on the Quality of English Language

Title: write "dual" instead of "duale"

Author Response

Dear reviewer, thanks again for your review. We are grateful for the two hints, we now removed the names of the authors (in the affiliation part) and corrected the word dual.

The new titel now is:

The dual role of Neutrophil Extracellular Traps (NETs) in
sepsis and ischemia-reperfusion injury: Comparative Analysis Across Murine Models

In addition, we are performing another english language check and will resubmit the new mansucript shortly.

Reviewer 2 Report

Comments and Suggestions for Authors

Authors satisfactory addressed all comments. I have suggestion authors can attempt for semi-quantitative measurement of IF experiment.

Author Response

Thank you very much for your recommendation. Thanks also for the comment regarding the semiquantiative statistics. We just really want to avoid performing any type of (semi-)quantitative statistics, when the underlying data for a solid calculation are not available, since that might lead to risky conclusions by the reader. Therefore we would prefer leaving it the way it is, although we share your thought and will improve this in our next project.

Reviewer 3 Report

Comments and Suggestions for Authors

Authors have addressed almost all the comments and concerns raised during the review process. The revisions made have improved of the article. However, authors should improve the words/sentence structure throughout the manuscript. For instance, Authors used NEXT (Line no. 188), it should be replaced with In addition/furthermore.

Author Response

Thanks for your review again. We are submitting an improved version of the english language of the manuscript, changes marked in yellow again.